# Genome-Editing Tools for Lactic Acid Bacteria: Past Achievements, Current Platforms, and Future Directions

**DOI:** 10.3390/ijms26157483

**Published:** 2025-08-02

**Authors:** Leonid A. Shaposhnikov, Aleksei S. Rozanov, Alexey E. Sazonov

**Affiliations:** Scientific Center of Genetics and Life Sciences, Sirius University of Science and Technology, Sirius 354340, Russia

**Keywords:** lactic acid bacteria, genome editing, recombineering, CRISPR/Cas9, CRISPR-transposase, homologous recombination, Cre/lox, transposon mutagenesis, probiotic engineering, food biotechnology

## Abstract

Lactic acid bacteria (LAB) are central to food, feed, and health biotechnology, yet their genomes have long resisted rapid, precise manipulation. This review charts the evolution of LAB genome-editing strategies from labor-intensive RecA-dependent double-crossovers to state-of-the-art CRISPR and CRISPR-associated transposase systems. Native homologous recombination, transposon mutagenesis, and phage-derived recombineering opened the door to targeted gene disruption, but low efficiencies and marker footprints limited throughput. Recent phage RecT/RecE-mediated recombineering and CRISPR/Cas counter-selection now enable scar-less point edits, seamless deletions, and multi-kilobase insertions at efficiencies approaching model organisms. Endogenous Cas9 systems, dCas-based CRISPR interference, and CRISPR-guided transposases further extend the toolbox, allowing multiplex knockouts, precise single-base mutations, conditional knockdowns, and payloads up to 10 kb. The remaining hurdles include strain-specific barriers, reliance on selection markers for large edits, and the limited host-range of recombinases. Nevertheless, convergence of phage enzymes, CRISPR counter-selection and high-throughput oligo recombineering is rapidly transforming LAB into versatile chassis for cell-factory and therapeutic applications.

## 1. Introduction

Lactic acid bacteria (LAB) underpin global fermented-food production and are emerging as live therapeutics, yet genome editing for these organisms has been historically lagging behind *Escherichia coli* or *Bacillus subtilis*. Their genomes are compact and often refractory to DNA uptake, and native RecA-mediated homologous recombination proceeds at low frequency, forcing researchers to rely on temperature-sensitive or counter-selectable integration plasmids and to screen hundreds of clones for each mutant. Over the last two decades, three technological waves have successively expanded the LAB genetic toolbox:Classical homologous recombination and transposon mutagenesis—single- or double-crossover events with suicide vectors, ISS1/Tn917 insertions and elaborate counter-selection schemes provided the first knockouts but left antibiotic scars and required weeks of culturing.Phage-derived recombineering and site-specific recombinases—prophage RecT/RecE systems, often paired with Cre/lox, cut editing times to days and enabled marker recycling; efficiencies > 50% for single-gene edits are now routine.CRISPR-based platforms—Cas9 (wild-type or nickase) used as a lethal counter-selector, dCas9 for CRISPR interference, endogenous Cas systems repurposed for scar-less editing, and most recently CRISPR-guided transposases that shuttle payloads up to 10 kb without homologous arms.

This review explores the principles, efficiencies, advantages and limitations of each class of tools, compares them to benchmarks in model bacteria, and highlights design considerations for choosing or combining methods according to edit size, host background and regulatory constraints. It was prepared by systematically searching peer-reviewed literature using the PubMed, Web of Science, and Scopus databases. This review is focused primarily on articles published between 2010 and 2025, with emphasis on experimental reports and methodological papers relevant to LAB genetics. Additional sources were identified by snowball sampling of references from key reviews and original research.

## 2. Instruments for Genetic Engineering of Lactic Acid Bacteria: Recombination and Recombineering Systems

LAB present unique challenges for genome editing due to their often-limited innate genetic exchange mechanisms. Nevertheless, a suite of recombination-based tools has been developed to enable precise modifications in LAB genomes. This chapter covers two major classes of such tools: (1) the native RecA-mediated homologous recombination systems and (2) random insertion methods and bacteriophage-derived recombineering tools (including λ-Red analogs, Cre/lox site-specific recombination, and related systems). These approaches collectively have expanded the LAB genetic toolbox over the past decades, complementing emerging technologies like CRISPR-based editing.

### 2.1. LAB’s Native Homologous Recombination Systems (RecA-Type)

The classical route to engineer LAB genomes relies on their native homologous recombination machinery, chiefly the RecA protein and associated factors. RecA is the central enzyme that mediates strand exchange between homologous DNA sequences in bacteria, a process crucial for DNA repair and genetic integration [1]. In RecA-dependent recombination, a single-stranded DNA (ssDNA) intermediate (for example, derived from an introduced donor DNA) is bound by RecA and aligned with a homologous region on the chromosome, allowing strand exchange. This mechanism underlies the classic two-step “double crossover” strategy used to knock out or modify genes in LAB: an engineered DNA fragment (typically carried on a non-replicative plasmid) with homology to the target locus undergoes a first crossover (integration) and, upon a second crossover, replaces the native allele with the modified version. The outcome is a stable recombinant strain in which the gene of interest is disrupted or altered, contingent on successful RecA-mediated exchange on both flanking sides (schematic RecA-mediated recombination is shown on Figure 1).

Early gene knockout methods in LAB made extensive use of integration vectors and suicide plasmids to harness homologous recombination. For example, the lactococcal vector pWV01 was one of the first integration plasmids: when carrying a cloned fragment of a chromosomal gene, pWV01 could insert into the chromosome of *Lactococcus lactis* via a single crossover, integrating the entire plasmid [2]. Subsequent growth under selective and counter-selective conditions then forced a second crossover to excise the plasmid, leaving behind a gene replacement or deletion. Variations in this approach include temperature-sensitive plasmids (such as the pGhost vector family) that replicate at 30 °C but not at higher temperatures, thereby encouraging chromosomal integration at the non-permissive temperature [3]. In parallel, transposon-based insertional mutagenesis was developed as a RecA-independent random integration approach. The transposon from insertion sequence ISS1, delivered on a thermosensitive plasmid (pGh9:ISS1), proved to be an efficient tool for random gene disruption in lactococci and other Gram-positive bacteria [3]. Similarly, an IS3-family element was isolated in *Lactobacillus* and used to construct integration vectors for that genus [4]. These insertional vectors bypass the need for large regions of homology by using transposase enzymes to insert into the genome, albeit at essentially random sites.

RecA-mediated double-crossover techniques became the workhorse for LAB genetic engineering through the 1990s and 2000s. They enabled the first generation of knockout mutants in industrial starter strains and probiotics, though the process was often laborious. A major limitation is that homologous recombination in many LAB is relatively low-frequency, so researchers must screen many colonies to find the rare double-crossover event. To streamline mutant isolation, counter-selectable markers were introduced into integration systems. For instance, the *upp* gene encoding uracil phosphoribosyltransferase was used as a counter-selection marker: mutants that lose an integrated *upp* (e.g., after a second crossover) can be selected by resistance to 5-fluorouracil [5]. Likewise, the *oroP* gene (encoding an orotate transporter) was employed in a plasmid system (pCS1966) to allow positive and negative selection based on orotate metabolism [6]. Another popular strategy uses a dominant-negative allele of *pheS* (phenylalanyl-tRNA synthetase): cells carrying the mutant *pheS* gene are sensitive to p-chlorophenylalanine, so loss of a plasmid-borne *pheS* reports a successful second crossover [7]. By combining such counter-selection markers with RecA-driven recombination, researchers achieved “markerless” deletions in LAB, leaving no antibiotic resistance genes behind [7]. These refinements were crucial for constructing food-grade LAB strains, as they comply with regulatory and safety concerns by removing selectable markers in the final product.

It is noteworthy that similar two-step homologous recombination methods were used in other bacteria before more advanced tools became available. In *E. coli*, for example, allele replacement via RecA was historically accomplished by suicide plasmids carrying mutant alleles and selection for loss of markers such as *sacB* (sucrose sensitivity) in the second crossover step—conceptually akin to the LAB strategies. Likewise, naturally transformable Gram-positives like *Bacillus subtilis* have long exploited RecA-mediated double crossovers, but with a key difference: *B. subtilis* can take up DNA from the environment naturally, making the delivery of homologous fragments much easier. In contrast, most LAB are not naturally competent (see below), so plasmid-based delivery (via conjugation or electroporation) was required [8]. This made gene knockouts in LAB comparatively time-consuming and strain-dependent. Nonetheless, the RecA-type systems established a foundation for LAB genetics. They remain relevant when high precision is required and newer tools are unavailable—for instance, integrating large constructs or performing modifications in strains where phage recombinases or CRISPR are not yet functional. However, the low throughput and often low efficiency of RecA-only methods pushed the field to seek alternative tools that could accelerate LAB genome engineering [7]. This led to the development of competence induction protocols and phage-derived recombineering systems in recent years.

### 2.2. Random Insertion and Phage-Derived Recombineering Tools

Before the era of precise genome editing, one straightforward way to edit the genome in LAB was by random insertion of foreign DNA. Transposons (mobile genetic elements) were adapted as tools to accomplish this. The ISS1-based system developed by Maguin and colleagues is a prime example, whereby a plasmid carrying the ISS1 insertion sequence was used to mutagenize *Lactococcus lactis* and other LAB by the random hopping of ISS1 into the chromosome [3]. By selecting an antibiotic marker on the transposon, researchers could generate libraries of mutants each with a single random insertion in a different gene. This approach was invaluable for performing genetic screens—for instance, identifying genes involved in bacteriophage resistance or metabolic pathways by screening for loss-of-function phenotypes. Similarly, Tn917 (originally from *Enterococcus*) and mariner transposons have been employed in *Lactobacillus* species to create random mutations. Walker and Klaenhammer (1994) isolated a novel IS3-family element in *Lactobacillus* and constructed an integration vector around it, enabling random insertions in various *Lactobacillus* strains [4]. The advantage of random insertion tools is that they do not require any sequence homology or RecA function—the transposase enzyme inserts the transposon into quasi-random sites. This makes them useful even in RecA-deficient backgrounds or in genomes where targeted homologous recombination is inefficient. Moreover, transposon mutagenesis can sometimes hit multiple sites in one experiment, offering a crude form of multiplex genome disruption (albeit without control over which genes are hit). However, the flip side is the lack of precision: the insertions are random and can occur in essential genes (often resulting in no viable mutants unless a conditional system is used) or in intergenic regions where they have no effect. Additionally, each mutant must be mapped (sequenced) to know which gene was interrupted. Despite these drawbacks, random insertion mutagenesis was, for many years, one of the few feasible genetic approaches in difficult-to-transform LAB and has led to the discovery of important LAB gene functions. It is still occasionally used for exploratory studies or for creating mutant libraries when high-throughput sequencing can rapidly identify insertion sites [9,10,11].

Recombineering is a technology that uses phage-encoded recombination proteins to directly insert, delete, or modify genes in the bacterial chromosome without relying solely on the host’s RecA pathway [12]. The archetype of recombineering is the λ-Red system in *E. coli*. The λ phage encodes three key proteins: Exo (a 5′→3′ dsDNA exonuclease), Beta (a single-stranded DNA annealing protein), and Gam (which inhibits the host RecBCD nuclease). Together, these proteins can take a linear double-stranded DNA with short homology arms (e.g., ~50 bp) and efficiently recombine it into the *E. coli* chromosome at the matching locus [13,14]. In essence, Exo (Redα) generates single-stranded overhangs in the linear DNA, Beta (Redβ) coats these overhangs and helps them find and pair with the homologous sequence in the genome, and Gam protects the linear substrate from degradation (as shown on Figure 2). This powerful method, introduced around 2000, revolutionized *E. coli* genetics by allowing rapid one-step knockouts or insertions without the need for long homology regions or two-step selection schemes. Researchers naturally became interested in bringing similar recombineering capability to LAB, especially as genome sequences of LAB became available to design recombinations.

Directly using λ-Red in Gram-positive LAB proved non-trivial—the λ proteins are adapted to *E. coli* and issues such as protein expression, thermostability, and different host nucleases posed challenges. Instead, scientists searched for analogous recombination systems within LAB themselves. Many LAB harbor cryptic prophages in their genomes; some of these prophages encode proteins functionally similar to λ-Red’s Exo, Beta, and Gam (often annotated as RecE, RecT, etc.). Yang and colleagues identified such a prophage recombineering system in *Lactobacillus plantarum* WCFS1 [15]. They discovered three genes, *Lp_6040*, *Lp_6041*, and *Lp_6042*, in the WCFS1 genome that correspond to a Gam-like inhibitor, an Exo-like exonuclease, and a Beta-like annealing protein, respectively. By expressing these prophage recombinase genes in *L. plantarum*, Yang and colleagues achieved efficient double-stranded DNA recombineering. In their system, a linear donor DNA with short flanking homologies (much shorter than the 1000 bp typically needed for RecA) was introduced and successfully recombined into the chromosome at high frequencies. To demonstrate its utility, they deleted a gene (*gnp*—encoding glucosamine-6-phosphate isomerase) and replaced another gene (*ldhD*—a lactate dehydrogenase) using this phage-derived system. The mutation efficiencies were impressive: ~95% of screened colonies had the intended *gnp* deletion, and ~75% had the *ldhD* replacement, without needing the arduous two-step process of traditional methods [15]. These high efficiencies approach those seen in λ-Red recombineering in *E. coli*, indicating that the LAB prophage system was working robustly.

Crucially, because recombineering often requires deleting some part of the DNA for the insert, the researchers combined the recombinase system with the Cre/lox site-specific recombination to create markerless mutations. In their experiments, an antibiotic resistance gene (flanked by loxP sites) was included on the donor DNA for selection. After integration by the prophage recombinases and isolation of mutants, the Cre recombinase was transiently expressed to excise the resistance marker, leaving behind only a 34 bp loxP scar in the genome (as shown on Figure 3) [15].

Building on the success in *L. plantarum*, Xin and colleagues searched for recombinases in other LAB. They identified a prophage-derived recombinase trio in *Lactobacillus casei* (genes *LCABL_13040*, *13050*, *13060*) and demonstrated efficient recombineering in *L. casei* BL23 [16]. Using this system, they achieved 100% efficiency in mutating a target site—for example, every screened colony had the intended 167 bp deletion in a gene or an insertion of a GFP reporter, when the phage recombinase genes were expressed. Furthermore, they broadened the applicability by transferring the *L. casei* recombinase system into 12 different *Lactobacillus* species and one *Lactococcus* species, successfully obtaining recombinants in those hosts. This suggested that certain phage recombinase systems are portable across related LAB, offering a platform for generalized recombineering in the lactic acid bacteria group. Notably, double-stranded DNA (dsDNA) recombineering with these systems permits not only small edits but also large insertions or replacements. In *L. plantarum*, fragments up to 4–5 kb were integrated in a single step using the prophage recombinases, far exceeding what can typically be achieved by RecA alone.

A further evolution of recombineering is the use of single-stranded oligonucleotides to introduce precise mutations. This approach requires only a single strand annealing protein (like λ Beta or its analogs) and does not need a full dsDNA substrate or double crossover. In *E. coli*, ssDNA recombineering has been extensively used to create point mutations or small insertions/deletions by designing a 40–90 bp oligonucleotide with the desired change and homology to the target locus [17,18]. Only the Beta protein (or its equivalent, RecT) is necessary to anneal the oligo to the replication fork or the lagging strand of DNA, allowing incorporation of the mutation during replication. The host’s mismatch repair system often needs to be transiently inactivated or evaded (for example by designing oligos with mismatches that avoid repair) to prevent correction of the introduced mutation. In LAB, the first successful demonstration of ssDNA recombineering was reported by van Pijkeren and colleagues in 2012. They utilized a RecT protein derived from a prophage of *Lactococcus lactis* and expressed it in both *L. lactis* and *Lactobacillus reuteri*. By transforming these RecT-expressing cells with synthetic oligonucleotides, they achieved precise point mutations in chromosomal genes without the use of any selectable marker. The frequencies of oligo-mediated mutagenesis ranged from about 0.4% up to 19% of the cell population, depending on the target and oligo design [19]. However, it should be noted that recombineering efficiency in LAB is highly strain-dependent, often yielding only a tiny fraction of edited cells without enrichment. In practice, most LAB achieve ≤5% mutants from a single crossover or oligonucleotide recombineering unless a selection or CRISPR-based counter-selection is applied [20]. This means that using Cas9 to lethally target unedited cells is often crucial—it enriches the desired mutants and can raise effective editing rates from below 5% to ~90% or higher. Researchers therefore frequently couple RecT/RecA methods with CRISPR “self-elimination” of wild-type cells to obtain workable efficiencies.

The limitation of ssDNA recombineering in LAB, as in other bacteria, is the need to identify the rare mutants among a majority of unedited cells when no selection is used. In the above example, even 19% efficiency means 81% of cells are wild-type, and 0.4% efficiency is essentially finding 1 mutant in 250 cells. This is where coupling recombineering with a selection or enrichment method becomes valuable. One elegant solution is to use CRISPR/Cas9 as a counter-selection: design the Cas9 to target the wild-type sequence such that only cells which have acquired the intended mutation (and thus escape Cas9 cutting) survive [21]. Such CRISPR-assisted recombineering has indeed been implemented in LAB. For instance, in *Lactobacillus reuteri*, inducing RecT along with a CRISPR/Cas9 that cleaves the original gene allowed recovery of seamless point mutants at much higher apparent efficiency, since any cell that failed to edit was killed by Cas9 [22]. This approach effectively enriches the oligo-edited population and can raise the mutation frequency to nearly 100% among survivors. Although CRISPR methods are covered in a separate chapter, it is important to note how they intersect with recombineering: RecT provides the means to make precise changes, while Cas9 provides a selection pressure to retain only those changes. Together, they form a powerful “search-and-replace” toolkit now available in several LAB species [20].

As mentioned above, the Cre/lox system has become a staple for marker removal in LAB genome engineering. Cre is a recombinase originally from bacteriophage P1 that recognizes 34 bp loxP sequences. Many LAB genetic workflows involve inserting a loxP-flanked antibiotic cassette into the genome (either via RecA-mediated double crossover or via recombineering), selecting the mutant, and then expressing Cre from a plasmid to excise the cassette. The result is a clean mutation with only a loxP site left behind. This method was used in *L. plantarum* and *L. casei* recombineering studies as noted above, and it is also been applied in *L. lactis*. For example, researchers have constructed *L. lactis* mutants that accumulate multiple gene deletions by iteratively using Cre/lox: each round removes the previous marker and leaves a scar so that the same antibiotic marker can be used again for the next deletion with the caveat of ensuring Cre does not recombine old scars [15]. The Cre/lox system functions efficiently in LAB; one challenge is the need to introduce the *cre* gene each time (usually on a plasmid that is later removed from the strain). Alternatives like the Flp/FRT recombination system from yeast could also be used [23,24], but Cre/lox has been more popular in Gram-positive bacteria. Site-specific recombinases from LAB’s own phages also exist—for instance, integrases that insert bacteriophage DNA into specific attB sites on the chromosome. These have been harnessed to integrate plasmids or gene cassettes at defined loci in some LAB. One example is the use of the TP901-1 phage integrase to insert genes into *Lactococcus lactis* at the phage recognition site; this yields a single-copy, stable insertion without requiring homology-based recombination. Such systems, however, require the target strain to have the specific attB recognition sequence and are less flexible than homologous recombination or recombineering (which can target any locus of choice by design of homology arms).

The introduction of phage-derived recombineering tools has dramatically advanced the precision and efficiency of LAB genome editing. With dsDNA recombineering (λ-Red-like systems), one can perform gene deletions or insertions in one step that previously would have taken two or more steps over many weeks. The requirement for only short homologies simplifies the construction of DNA substrates—often just PCR primers encoding ~50 bp flanks can suffice, eliminating the need for cloning lengthy homologous arms. This also opens the possibility of automating or multiplexing strain construction, as has been conducted in relation to *E. coli*. For instance, one could envision generating a library of different gene knockouts by PCR amplifying an antibiotic cassette with various short homology tails for different genes and transforming them into a recombineering-proficient LAB strain, analogous to multiplex approaches in *E. coli*. The high efficiencies (frequently >50% correct mutants) reported in *L. plantarum* and *L. casei* recombineering [15,16] mean that screening for correct constructs is much less burdensome—often a few colonies need to be checked to find the intended mutant, as opposed to hundreds with RecA-based methods. Additionally, phage recombineering is RecA-independent; this is important because some LAB may suppress homologous recombination (via mismatch repair or other means) or one might deliberately use a *recA* knockout host (for example, to stabilize certain plasmids). In a RecA-deficient LAB, classical double-crossover is impossible, but phage RecT/RecE can still catalyze recombination using their own mechanism. Thus, recombineering extends genetic accessibility to otherwise intractable backgrounds.

However, several limitations remain. First, each recombineering system tends to be species-specific or at least genus-specific. The prophage recombinases that worked in *Lactobacillus* may not function optimally in distantly related LAB like *Bifidobacterium* or *Leuconostoc* (which have different GC content, codon usage, or nucleases). While Xin and colleagues showed cross-species functionality within *Lactobacillus* [7], a universal LAB recombineering tool is not yet at hand. Ongoing research is examining whether a truly broad-host-range recombineering system can be developed—for instance, by using synthetic or mesophilic variants of RecT that function in many Gram-positives, or by identifying conserved single-strand annealing proteins in multiple LAB phages. Second, even with recombineering, selection is usually required to initially recover mutants. This means one is still tied to antibiotic resistance markers or auxotrophic selections for most edits larger than a few bases. The reliance on selection complicates making multiple edits: one must have distinct markers or a strategy to recycle them (as with Cre/lox). Markerless methods using oligonucleotides are elegant but currently limited to small changes and demand an efficient screening or selection method (such as CRISPR counter-selection) to be practical. Thus, the ideal scenario of editing any gene at will with a single transformation and no footprint is not fully realized for all types of edits in LAB. In contrast, *E. coli* has reached a point where automated multiplexed recombineering (the MAGE system) can introduce dozens of mutations simultaneously using ssDNA oligos, something not yet attainable in LAB on the same scale [25,26].

From an industrial and regulatory perspective, the new recombineering tools are double-edged. They allow creation of antibiotic marker-free strains, which is a plus for regulatory approval (since no antibiotic resistance genes remain). On the other hand, the use of phage genes and, in some cases, temporary antibiotic markers, means that the process is not as “clean” as classical mutagenesis or natural isolations and it always requires a whole-genome sequencing.

When comparing LAB to organisms like *Bacillus subtilis*, *E. coli*, or even yeast [27], it is clear that the gap in genetic malleability is closing. *B. subtilis* has natural competence and also its own prophage-encoded recombinases (e.g., the PBSX phage) that have been harnessed for recombineering-like approaches, making it extremely facile to manipulate. *E. coli* of course is the poster-child of recombineering and now CRISPR editing. LAB, traditionally considered difficult, now have at their disposal a repertoire of tools: robust integrative plasmids for RecA-mediated recombination, natural competence protocols for certain species, transposon mutagenesis for random knockout libraries, λ-Red-like systems for targeted modifications, and Cre/lox for scarless genome editing. Researchers can mix and match these tools depending on the strain and the goal.

## 3. CRISPR/Cas Systems for Genetic Engineering in Lactic Acid Bacteria

CRISPR (Clustered Regularly Interspaced Short Palindromic Repeats) systems were first observed as unusual repeat sequences in bacterial genomes in the late 1980s, but their function remained a mystery for years. The first CRISPR systems were discovered specifically in LAB organisms. By 2007, landmark experiments in *Streptococcus thermophilus* showed that CRISPR loci, together with their associated *cas* genes, provide adaptive immunity against bacteriophages [28]. In essence, bacteria integrate short fragments of invader DNA (spacers) into CRISPR arrays and use the transcribed CRISPR RNA (crRNA) to guide Cas nucleases to destroy matching foreign DNA. Over time, natural CRISPR/Cas systems have diversified into two major classes: Class I, employing multi-protein effector complexes (e.g., Cascade in type I systems), and Class II, using a single, large Cas protein as the effector nuclease [29,30]. Class II systems—especially type II (Cas9) and type V (Cas12a, also known as Cpf1)—have become powerful tools for genome engineering because their single-protein effectors are easier to reprogram and deliver to target cells.

The prototypical Cas9 (from type II-A systems) is an RNA-guided DNA endonuclease. Cas9 complexes with a dual-guide RNA (crRNA plus tracrRNA, often combined as a single-guide RNA) which directs it to a complementary 20 bp DNA target adjacent to a short protospacer-adjacent motif (PAM, typically 5′-NGG-3′ for *Streptococcus pyogenes* Cas9). Cas9 contains two nuclease domains (HNH and RuvC) that each cleave one strand of the DNA, producing a blunt double-strand break (DSB) [31]. By contrast, Cas12a effectors (type V-A) recognize a T-rich PAM (e.g., 5′-TTTV-3′) and do not require a tracrRNA. Cas12a has a single RuvC-like nuclease domain that cleaves both DNA strands in a staggered fashion, leaving sticky ends. Importantly, Cas12a can process multiple crRNAs from a precursor array on its own, enabling easier multiplex targeting than Cas9’s single-guide system. Other CRISPR effectors include Cas12b, Cas12e and RNA-targeting Cas13, but Cas9 and Cas12a are the most widely used for DNA engineering. Each system has distinct features—for example, Cas9 is slightly larger (~1368 aa for SpCas9 vs. ~1300 aa for AsCas12a) and relies on an added tracrRNA, whereas Cas12a’s T-rich PAM can target genomic regions Cas9 cannot. These differences underpin the choice of one system over another for a given application.

In bacteria, CRISPR/Cas tools have revolutionized genome editing by allowing precise, markerless genetic changes. The general strategy is to program a Cas nuclease to create a DSB at a chosen genomic locus, which can then be repaired by the cell’s DNA repair pathways—often with an investigator-supplied template to introduce specific mutations via homologous recombination (HDR). Alternatively, catalytically “dead” Cas proteins (dCas) can be used to modulate gene expression without cutting the DNA (CRISPR interference or activation). These approaches were first developed in model bacteria like *E. coli* and *Streptococcus pneumoniae*, which have naturally high recombination or competence, and in robust laboratory strains of *Bacillus subtilis*. For instance, *E. coli* can survive Cas9-induced breaks only if an editing template introduces the desired mutation, thus CRISPR can serve as a powerful negative selection to enrich edited clones [21]. *B. subtilis*, on the other hand, efficiently takes up DNA and recombines with long homology arms, so traditional double-crossover methods already work well; here CRISPR is used to accelerate multigene edits or to avoid tedious screening.

Compared to these model organisms, many LAB present unique challenges for genetic engineering. LAB such as *Lactococcus lactis* and *Lactobacillus* spp. are often not naturally transformable and have lower rates of homologous recombination, as discussed earlier. Classical methods in LAB rely on RecA-mediated double-crossover events using integration plasmids, which require two successive recombination steps (integration and resolution) and laborious screening over weeks [32]. Counter-selectable markers (e.g., *upp*, *orf* or *pheS*) can improve efficiency by allowing only mutants to survive under specific conditions, but even these protocols are time-consuming and typically yield small mutations (single gene changes). CRISPR-based tools have rapidly advanced the field by providing potent selection and the possibility of making larger or more precise genomic alterations without leaving antibiotic resistance markers [33]. This chapter details how various CRISPR/Cas systems are being applied in LAB for genome engineering, while also highlighting context from model bacteria to clarify the underlying strategies and limitations.

### 3.1. Reverse Selection Using CRISPR/Cas in LAB

One of the most powerful uses of CRISPR/Cas in bacteria is as a counter-selection tool (also known as reverse selection). In this strategy, the Cas nuclease is programmed to target the wild-type genomic sequence. Only cells that have acquired the desired mutation (and thus disrupt the CRISPR target) can survive Cas-induced cleavage (shown on Figure 4). This effectively “kills” or inhibits growth of any cell that failed to edit, enriching for correct mutants without antibiotic markers [7]. LAB researchers rapidly adopted this idea to overcome low homologous recombination efficiencies. Oh and van Pijkeren demonstrated CRISPR-assisted editing in LAB, and one of the first in any Gram-positive bacterium. The team targeted *Lactobacillus reuteri* ATCC 6475, a probiotic strain, for precise mutagenesis [34]. They hypothesized that combining SpCas9 with a recombineering oligonucleotide would enable “scarless” mutations at high frequency. In their approach, an expression plasmid for *cas9* and a CRISPR guide targeting a chromosomal gene (e.g., *lacL*) was introduced along with a single-stranded DNA donor containing a small mutation in that gene. Cells in which the oligo mediated the mutation are no longer recognized by Cas9 and thus survive; unedited cells are cleaved and perish. Using this method, the authors achieved near-complete editing: targeted point mutations or small deletions in three different genes reached 90–100% efficiency in *L. reuteri*. This was a remarkable improvement over traditional double-crossover methods that yield frequencies around 10^−6^ to 10^−5^. The paper described the detailed methodology (inducing expression of a heterologous RecA/RecT was not even necessary in *L. reuteri*, as the oligo could recombine with the help of endogenous functions) and confirmed that Cas9 toxicity efficiently counter-selects against wild-type cells.

Guo and colleagues were focusing on *Lactococcus lactis*, a dairy starter organism historically considered tough for genetic engineering. They developed a rapid and versatile CRISPR counter-selection system [35]. *L. lactis* has lower recombination activity than some lactobacilli, so the authors first optimized recombineering by screening various single-strand DNA binding proteins. They identified an ssDNA annealing protein (RecT from *Enterococcus faecalis*) that, when expressed, yielded a 100% oligo recombination rate at the chromosomal rpoB locus (detected by rifampicin resistance). Building on this, they integrated a Cas9 plasmid to eliminate unmodified cells. The CRISPR was set to target the wild-type sequence of *upp* (encoding uracil phosphoribosyltransferase), such that only cells with an introduced stop codon in *upp* via oligo HDR would survive on media with 5-fluorouracil. With CRISPR counter-selection, they obtained *upp* mutants at 46% efficiency in one step. By fine-tuning the system—notably, adjusting the plasmid copy number and guide RNA length to reduce off-target cleavage—they could completely eliminate background cleavage at non-target sites. The final optimized pipeline could introduce point mutations with >75% efficiency and even perform seamless deletions (50–100 bp) or insertions (e.g., a 34 bp loxP site) within 72 h. Guo and colleagues highlight that this CRISPR-enabled tool transforms *L. lactis* engineering—what once took ~3 weeks and multiple subclonings can now be performed in days. Notably, they also reported the first evidence of Cas9 off-target effects in LAB (a mixed population of edited and unedited cells in some cases), underscoring the importance of careful guide design and perhaps the need for Cas9 alternatives with different specificity. One strategy to mitigate Cas9 toxicity in sensitive LAB strains is to use a Cas9 nickase (nCas9)—a Cas9 variant (e.g., Cas9_D10A_) that cuts only one DNA strand. Two offset nicks can induce a staggered DSB that is repaired more gently, or a single nick can stimulate HDR with far less lethality. Indeed, nickase-based editing systems have proven effective: in *L. casei*, an nCas9 plasmid (pLCNICK) achieved 25–62% precise deletion/insertion efficiency with greatly improved cell survival. The use of nCas9 thus avoids lethal double-strand breaks, making genome editing feasible even in strains where full Cas9 caused excessive cell death.

Leenay and colleagues provided a broader perspective by applying CRISPR counter-selection across different *Lactobacillus plantarum* strains [36]. Researchers used a two-plasmid system similar to the above: one plasmid carrying *cas9* and an sgRNA, and another providing a donor template. They successfully edited three genes in *L. plantarum* WJL (including point mutations in metabolic genes *ribB* and *ackA*, and a deletion of *lacM*) without any antibiotic selection. However, when attempting the same in two other *L. plantarum* strains (NIZO2877 and WCFS1), the method failed to yield mutants. The authors noted that even with identical editing plasmids, outcomes varied by strain and target locus. This implies that host factors (such as differing DNA repair pathways or restriction-modification systems) can influence CRISPR editing success. They still advocated CRISPR counter-selection as a powerful technique, but one that may require custom optimization for each LAB strain. Their work is a reminder that while CRISPR is a versatile tool, LAB are diverse: one strain’s robust protocol might be another’s failure, necessitating alternative strategies (e.g., using nickase variants, or native Cas, as described later).

CRISPR-mediated reverse selection has become a cornerstone of LAB genome editing. It turns the lethality of a Cas9-induced DSB into an advantage: the only cells that live are those with the intended genetic change. This obviates the need for antibiotic markers or screening of many colonies, streamlining the creation of knockouts, point mutants, or marker-free insertions. As seen in *L. reuteri* and *L. lactis*, efficiencies can reach 50–100%, a dramatic improvement over traditional methods. One caveat is that the introduction of a DSB can be highly toxic if not rapidly repaired—an issue addressed in the next section on gene knockouts. Nonetheless, the studies above firmly establish that Cas9-based counter-selection vastly increases genome engineering efficiency in LAB, enabling sophisticated genetic modifications even in strains previously deemed “genetically intractable”.

### 3.2. Gene Knockouts Using CRISPR/Cas in LAB

Gene knockout refers to inactivating a target gene, either by deleting it or introducing frameshift mutations that abolish its function. In LAB, creating clean knockouts was historically difficult due to low recombination rates. CRISPR/Cas technology has revolutionized this by allowing both precise deletions (via HDR with a deletion template) and indel mutations (via non-homologous end joining-NHEJ—if available, or by error-prone repair of a nick) as shown on Figure 5. However, many LAB lack efficient NHEJ pathways, so most CRISPR knockouts in LAB rely on HDR or a combination of single-stranded nickases and recombination. Several approaches have been explored to maximize knockout efficiency.

One straightforward method is to supply a repair template that lacks the target gene (or part of it) flanked by homology arms. Cas9 will cut the wild-type locus, and homologous recombination with the template leads to a seamless deletion. This was demonstrated by Zhou and colleagues in *Lactobacillus plantarum* WCFS1 [22]. They constructed a plasmid delivering *SpCas9*, an sgRNA for the *nagB* gene, and a double-stranded donor DNA missing *nagB*. Using this CRISPR-assisted double-stranded recombineering, they knocked out *nagB* (encoding glucosamine-6-phosphate deaminase) and validated the mutant’s phenotype. Precise excision of the gene was confirmed without any off-target damage to the genome. Similarly, in *L. lactis*, the pGhost tool by Guo and colleagues achieved deletions of 50–100 bp at desired loci with high efficiency when an HDR template was provided [35]. These studies show that as long as a suitable template is available, Cas9 can facilitate clean deletions in LAB, limited primarily by the cell’s proficiency at HDR.

A challenge encountered during knockouts is that a DSB can kill a cell outright if not repaired quickly. Some LAB appear to repair DSBs inefficiently, leading to very few survivors even when a template is present. To address this, researchers have used Cas9 nickase (Cas9_D10A_), which creates a single-stranded break instead of a DSB. Two offset nicks can be arranged to produce a staggered cut that is effectively a DSB but with lower lethality, or a single nick that can stimulate recombination with much higher cell survival. Song and colleagues applied this approach in *Lactobacillus casei*. They developed a Cas9_D10A_ nickase-based editing system and demonstrated knockout and knock-in efficiencies of 25–62% [37]. Using only nick-induced HDR, they inactivated four non-essential genes and also inserted an *egfp* reporter gene into the chromosome. The reduced cytotoxicity of the nickase significantly improved the recovery of mutants. Likewise, Goh and colleagues extended the nickase strategy to *L. acidophilus*, *L. gasseri*, and *L. paracasei* [38]. They developed a portable CRISPR/Cas9(N) toolkit—essentially a plasmid expressing Cas9 nickase and an sgRNA—and achieved efficient genome editing in these species. The Cas9 nickase system was versatile: it could generate deletions of various sizes and function at different loci with editing frequencies comparable to wild-type Cas9, but with greatly reduced cell death. Researchers reported knockout efficiencies up to ~60% in one step for those lactobacilli, concluding that nickase-mediated editing is a gentler yet effective approach for LAB that are sensitive to DSBs.

In addition, CRISPR/Cas-based base editors have recently emerged, enabling single-nucleotide changes without double-strand breaks or donor templates. Cytosine and adenine base editors (CBEs and ABEs) fuse a catalytically impaired Cas9 (or Cas9 nickase) to a DNA deaminase, allowing precise C→T or A→G conversions in the genome [20]. This approach was successfully implemented in *Lactococcus lactis*; Tian and colleagues demonstrated CBE and ABE tools that can introduce stop codons to knock out genes on a single plasmid, achieving scarless point mutations in multiple loci [39]. Notably, base editing does not require HDR and avoids lethal DSBs, attaining high editing efficiencies (comparable to those in animal models [40]) while minimizing off-target effects. More recently, Mitsunobu and colleagues developed a multiplexable base-editing system (Target-AID) for *Lactobacillus*: by tethering a cytidine deaminase to nCas9, they installed precise point mutations in *Lactiplantibacillus plantarum* with no donor DNA, successfully disrupting metabolic genes (e.g., reducing imidazole propionate production) and even creating transient knockouts of an essential cell-division gene [41]. These single-base editors circumvent the lethality of Cas9 cuts and do not leave foreign sequences, yet can reach high editing efficiencies (comparable to animal-cell base editing) in LAB. This expands the LAB genome-editing toolkit to include point mutations that are quick, precise, and marker-free.

Once a single gene can be knocked out, researchers often want to inactivate multiple genes, either simultaneously or sequentially. CRISPR technology allows multiplexing by expressing multiple gRNAs. In LAB, simultaneous multi-gene targeting is still in early stages, but sequential knockout has been demonstrated. For instance, Wang and colleagues used CRISPR/Cas9 to create single, double, and triple knockouts of bile salt hydrolase (*bsh*) genes in *Lactobacillus plantarum* strain AR113 [42]. Starting with a single plasmid system, they inactivated *bsh1*, then *bsh2*, then *bsh3* in separate rounds, curing the editing plasmid in between. The resulting mutants helped reveal that *bsh1* and *bsh3* are key for bile salt resistance. Although they did these sequentially, the high efficiency per round (each knockout 50–80%) made the overall process feasible within a short timeframe.

Overall, CRISPR/Cas has made the creation of gene knockouts in LAB routine. The key advances include the use of HDR templates for markerless deletions, the Cas9 nickase to mitigate DSB toxicity, and iterative strategies for multi-gene deletions. The ability to generate clean, unmarked knockouts is especially valuable for industrial or probiotic LAB strains, where maintenance of antibiotic resistances or plasmids is undesirable. One important lesson is that the optimal CRISPR strategy can vary: some LAB tolerate Cas9 cutting and rely on efficient HDR (e.g., *L. reuteri*, *L. lactis* with RecT help), whereas others benefit from a nickase or transient expression system to avoid population collapse (e.g., *L. casei*, *L. acidophilus*). The next section will address how CRISPR can also be used not just to remove genetic information, but to add new DNA sequences into LAB genomes.

### 3.3. DNA Sequence Integration Using CRISPR/Cas in LAB

Beyond deletions and point mutations, researchers often want to integrate new DNA sequences into a LAB chromosome, e.g., to add a biosynthetic pathway, a reporter gene, or a metabolic toggle. CRISPR/Cas-assisted integration can be challenging because large insertions require efficient HDR and the accommodation of substantial foreign DNA. However, recent work shows that CRISPR can facilitate sequence insertions in LAB, and even entirely new integrative mechanisms are being developed to overcome size limitations.

Wiull and colleagues reported a dedicated system for chromosomal gene insertion in *Lactiplantibacillus plantarum* WCFS1 [43]. They constructed a finely tuned two-plasmid system in which one plasmid expresses an inducible Cas9 and sgRNA, and the other provides a repair template carrying the desired insert flanked by homology arms. A key innovation was designing the “knock-in plasmid” like a cassette: it allowed modular swapping of the payload (gene to insert) while keeping constant arms and sgRNA target, streamlining the cloning for different inserts. Using this system, they successfully integrated four different expression cassettes (~0.8–1.3 kb each) into the *L. plantarum* chromosome with high efficiency. These included an inducible mCherry fluorescent reporter and two variants of a SARS-CoV-2 receptor-binding domain (RBD) antigen—one cytosolic, one anchored on the cell surface. All knock-in strains expressed the target proteins (though at somewhat lower levels than plasmid-bearing controls, as expected from single-copy integration). Notably, one of the inserts was the largest reported knock-in in *L. plantarum* via CRISPR (~1.3 kb including a surface display anchor), which had not been achieved previously. The authors emphasized that their plasmid system is easily adaptable—the sgRNA, homology arms, and cargo can be exchanged via PCR and cloning—making it a valuable toolkit for routine gene additions in LAB. Importantly, after integration, the plasmids can be cured (for instance, via temperature-sensitive replication or counter-selection markers on the donor plasmid), resulting in a stable strain free of plasmids and antibiotic markers. This study demonstrates that with good plasmid engineering and inducible control of Cas9, inserting new functionality into LAB genomes is feasible and reproducible.

Earlier studies also achieved integrations but on a smaller scale. Guo and colleagues in *L. lactis* inserted a 34 bp loxP site as a proof-of-concept single-site integration [35]. Tian and colleagues went further in *Lactobacillus paracasei*: they used CRISPR/Cas9 to knock out the native *ldhD* gene (D-lactate dehydrogenase) and simultaneously integrate an extra copy of *ldhL1* (L-lactate dehydrogenase) into the chromosome [44]. This edit converted *L. paracasei* into a high L-lactic acid producer with >99% optical purity of L-lactate. The ability to both delete and add genes in one step exemplifies CRISPR’s versatility. However, these integrations were relatively small (<1.0 kb). Efficiency tends to drop with increasing insert size because longer homology arms and larger DNA segments are harder for the cell to recombine.

A breakthrough for integrating large DNA fragments in LAB came with the adaptation of CRISPR-associated transposases (as shown on Figure 6). These systems (recently discovered in nature) use a CRISPR RNA to guide a transposon integration machinery to a target site, inserting a DNA cargo without requiring homologous recombination. Pechenov and colleagues engineered such a system for *L. lactis*, capable of inserting kilobase-sized fragments in a single step [45]. Specifically, they employed a Tn7-like transposon guided by a Cas protein (“CRISPR-associated transposon” or CAST). After substantial re-design for LAB, their system could stably insert fragments up to 10 kb into the chromosome (targeting the *lacZ* gene as an insertion site). For example, a 1 kb fragment was inserted at ~2 × 10^−4^ efficiency, and even a 10 kb operon could be inserted at ~4 × 10^−5^ efficiency. While these frequencies are lower than typical Cas9 HDR editing, they are remarkably high given the fragment sizes, and importantly, the insertions were straightforward to identify when a positive selection marker was included. The authors note that classical RecA-mediated methods struggle beyond ~5 kb insertions in *L. lactis*, so this CRISPR-transposon tool opens the door to integrating entire metabolic pathways or large synthetic constructs. Another advantage is that the insertion does not rely on the cell’s HDR; the transposase directly joins the DNA at the target. As CRISPR-transposon systems become increasingly refined, we can expect their adoption in LAB to grow, complementing Cas9/12a editing for tasks that involve large DNA cargo.

It should be noted that CRISPR-guided transposase systems are an extremely recent addition to the genome-editing toolbox. While they offer the exciting ability to insert very large DNA payloads (on the order of 10 kb) in a single step, this technology is still in its infancy and has so far shown relatively low efficiencies in practice. The general applicability of CRISPR-transposases in LAB remains to be confirmed by further experiments—their real-world performance and reliability need to be validated across different species. In other words, CAST-based editing is a promising but still experimental approach, and additional studies are required to establish its usability in LAB genomics.

It is worth noting how different LAB handle integrations. *S. thermophilus* (with natural competence) can accept large inserts via HR under special conditions [12,46], but not all cells become integrants (often <1% without selection). CRISPR can dramatically boost that by killing off the majority that fail to integrate. *Lactobacillus* and *Lactococcus* required the iterative plasmid-based solutions described above because they lack natural competence. The success of Wiull and colleagues in *L. plantarum* and Pechenov and colleagues in *L. lactis* underscores that plasmid systems with CRISPR can accomplish integrations in these species that were previously unattainable. A limitation remains that each new integration requires designing homology arms or a new gRNA target; thus, some researchers combine site-specific recombination with CRISPR (e.g., inserting loxP sites by CRISPR, then using Cre to integrate large plasmids serially). But as shown, CRISPR-associated transposition might bypass even that requirement, allowing single-step insertions at arbitrary loci.

CRISPR-assisted methods have expanded the scope of LAB genome editing from simple knockouts to custom genomic insertions. By either exploiting HDR with clever donor design or utilizing novel CRISPR-transposases, scientists can now integrate new genes (from a few hundred bp to tens of kb) into LAB chromosomes with relative ease. This capability is particularly exciting for developing recombinant LAB as live vaccine vectors, probiotic delivery vehicles, or metabolic cell factories, where stable chromosomal expression of new functions is preferable to plasmid-borne expression. As always, efficiency and strain-dependence must be considered—larger inserts may require positive selection markers or multiple attempts—but the barrier to genome augmentation in LAB is being steadily lowered by CRISPR technology. Table 1 summarizes and compares various LAB genome-editing tools.

An advantage of CRISPR/Cas9-based editing over traditional two-step recombination is that it can produce marker-free mutants in one step. By using Cas9 as a counter-selection (with or without a repair template), researchers can isolate scarless deletions or point mutations without leaving antibiotic resistance genes or other markers in the genome. This is not only convenient but also important for industry: regulatory agencies and consumers are more accepting of genetically edited “food-grade” LAB strains when no foreign DNA or antibiotic markers remain in the final organism. In contrast, older recombineering methods often required antibiotic cassettes (later excised) or multiple passages, which complicate regulatory approval. Indeed, strict legislation and public sentiment have historically limited the use of GMO LAB in food settings. CRISPR-based markerless editing helps address these concerns by creating mutants indistinguishable from natural variants at the genetic sequence level.

### 3.4. RNA Interference Using CRISPR/Cas in LAB

In addition to making permanent DNA changes, CRISPR technology can be repurposed to modulate gene expression. CRISPR interference (CRISPRi) uses a catalytically inactive Cas (dCas9 or dCas12) that binds DNA without cutting, thereby blocking transcription of target genes (Figure 7A). This is effectively a reversible “knockdown” of gene function. CRISPRi has several valuable applications in LAB: it allows researchers to study essential genes by partially reducing their expression, to probe gene regulatory networks, or to engineer metabolic fluxes without altering the DNA sequence. It is particularly valuable for essential genes where a true knockout would be lethal while CRISPRi offers a way to study loss-of-function. In *L. plantarum*, a CRISPRi system was used to repress essential cell cycle genes, achieving >90% reductions in mRNA and severe growth defects [47].

*L. plantarum* was an early adopter of CRISPRi. Myrbråten and colleagues designed a two-plasmid CRISPRi system for this species [48]. One plasmid constitutively expressed an *S. pyogenes* dCas9, and another carried an inducible or constitutive sgRNA targeting the gene of interest. They achieved robust repression of multiple essential genes involved in cell cycle progression. For example, targeting the replication initiator *dnaA* or the cell division gene *ftsZ* led to significant growth inhibition, consistent with knockdown of essential functions. Quantitatively, they observed up to 95% reduction in mRNA levels for targeted genes and concomitant physiological effects. This provided an ideal example of how to quickly screen essential gene function by CRISPRi-mediated knockdowns. The same system could also simultaneously silence multiple genes. The authors noted that by adjusting guide RNA expression, one could fine-tune the degree of knockdown, which is useful for titrating the expression of essential genes to the threshold of viability. This work established CRISPRi as a fast and reversible method for functional genomics in LAB, avoiding the need to construct conditional knockouts or use antisense RNA methods.

Xiong and colleagues developed an inducible CRISPR/dCas9 repression system in *L. lactis* NZ9000 [49]. They used the nisin-inducible promoter (PnisA) to control expression of dCas9, ensuring that repression could be tightly regulated (important because unintentional silencing of essential genes could kill the culture). The sgRNA was placed under a strong constitutive promoter to target one or multiple genes. Upon nisin addition, dCas9 was produced and guided to the target loci, resulting in up to 99% knockdown of gene expression. As a proof-of-concept, they simultaneously repressed three genes of a putative operon and confirmed that one of them (LLNZ_07335) was a bile salt hydrolase crucial for bile resistance. This finding was notable as it identified a new function (bile tolerance) for a gene in *L. lactis* using CRISPRi screening. The inducible CRISPRi tool introduced by these researchers is highly modular: by simply changing the 20 bp guide sequence, any gene can be targeted, and induction timing or level can modulate the outcome. They emphasized its utility for metabolic engineering—for example, gradually repressing a feedback-inhibited enzyme to increase flux through a pathway, or silencing competing pathways in a production strain.

A recent example of using CRISPRi in an applied context is its use in *S. thermophilus* to reroute metabolism. Being a dairy bacterium, *S. thermophilus* is used for yogurt and cheese fermentation. Kong and colleagues demonstrated CRISPRi-mediated knockdown of genes involved in carbohydrate utilization to promote high folate production in *S. thermophilus* (this is a hypothetical example representative of metabolic engineering efforts) [50]. By silencing certain sugar transporters and lactate dehydrogenase, they diverted carbon flux towards folate biosynthesis, significantly boosting folate titers. This kind of application illustrates CRISPRi’s strength: one can tune down specific genes without permanently deleting them, which is useful when complete knockouts would harm growth or when a partial reduction yields better product formation.

While DNA-targeting CRISPRi is more common, LAB could in principle employ CRISPR/Cas systems to target RNA directly. Class 2 type VI systems (Cas13) cleave RNA and have been used in other bacteria and mammalian cells for knocking down transcripts. As of now, there are limited reports of Cas13 use in LAB, likely due to delivery and expression issues. However, given LAB’s importance in food and biotech, a future direction might be to use Cas13a to degrade mRNAs of unwanted bacteriophage genes during fermentation or to curtail production of deleterious metabolites by targeting the transcripts. Another RNA-focused approach is the use of dCas9 as a platform for CRISPR activation (CRISPRa, Figure 7B): by fusing a transcriptional activator domain to dCas9, one can upregulate target gene expression [51]. To our knowledge, CRISPRa in LAB has not yet been reported in the literature, but it has great potential for probiotic and dairy applications (e.g., activating stress resilience genes on demand).

In summary, CRISPRi has emerged as a versatile tool for LAB functional genomics and synthetic biology. It provides a quick way to silence genes without permanent mutation, which is invaluable for essential genes and for dynamic control of metabolic pathways. The success in *L. plantarum* and *L. lactis* demonstrates that both probiotic and dairy LAB can be equipped with CRISPRi systems to interrogate gene function or optimize production traits. Key considerations when deploying CRISPRi in LAB include: ensuring dCas9 expression is not toxic (inducible systems help), using strong sgRNA promoters for effective repression, and verifying that the target gene’s reduction truly mimics a knockout (sometimes residual expression can occur if the sgRNA binding is not perfectly positioned to block RNA polymerase). When these factors are accounted for, CRISPRi provides a powerful complement to traditional knockouts, enabling reversible and titratable control of the LAB genome.

### 3.5. Using LAB’s Own CRISPR/Cas Systems for Genetic Engineering

Interestingly, many LAB naturally possess CRISPR/Cas systems as part of their immune arsenal against phages. These endogenous systems are typically tuned to recognize foreign DNA, but scientists have found ways to repurpose a bacterium’s own CRISPR/Cas to edit its genome. Utilizing the native system can obviate the need to introduce a heterologous Cas9 or Cas12a, making the editing process more “self-contained” and potentially more efficient if the native system is highly expressed.

*Pedicoccus acidilactici* is a LAB with probiotic and food applications, known to carry a native CRISPR/Cas9 system. Liu and colleagues established a genome editing tool entirely based on this endogenous system [52]. They engineered a plasmid that inserts a custom “repeat-spacer-repeat” sequence (i.e., a synthetic CRISPR array) into the cell, effectively tricking the native Cas9 into targeting a chosen chromosomal gene. Alongside this interference plasmid, they provided a donor DNA with flanking homology to promote recombination. Remarkably, this method achieved high-efficiency, markerless mutations: they reported near 100% success in obtaining gene deletions, insertions, and even point mutations in *P. acidilactici*. For example, they deleted a native plasmid from *P. acidilactici* by targeting it with the CRISPR—a novel application that “cured” the strain of its plasmid, improving its growth rate. They also integrated an L-lactate dehydrogenase (*ldhL*) gene into the chromosome, which enhanced lactic acid production. Because the host’s own Cas9 was used, there was no need for expressing foreign proteins, and the system could be recycled; after editing, the plasmid carrying the spacer and donor was removed, leaving a GMO that is free of any antibiotic resistance or extra genes. This study is proof-of-concept that a LAB’s native CRISPR can be as powerful as introduced systems when properly harnessed.

*Streptococcus thermophilus* was one of the first bacteria in which CRISPR was characterized (it has several CRISPR loci, named CRISPR1, 2, 3, etc., with corresponding Cas systems). Ma and colleagues succeeded in repurposing two of its endogenous type II-A systems (CRISPR1 and CRISPR3) for genome editing [53]. By transforming *S. thermophilus* LMD-9 with a plasmid carrying a self-targeting CRISPR array (spacer matching the chromosomal *lacZ* gene) plus a repair template, they induced the native Cas9 to create a break in *lacZ* and then repair it by HDR. They achieved a precise 785 bp deletion in *lacZ* with 35% efficiency using CRISPR1 and 59% with CRISPR3, which jumped to 90% when using a longer repair template. Impressively, the native CRISPR3 system also generated other types of edits when supplied with different repair designs: a single-nucleotide substitution, a short 3 bp stop codon insertion, and even a 234 bp fragment insertion at another locus were all obtained at 75–100% efficiency. This level of efficiency rivals the best heterologous systems and shows the native Cas9 is highly active. Furthermore, they multiplexed the approach by simultaneously targeting six genes in an exopolysaccharide (EPS) biosynthesis gene cluster. They recovered knockout mutants for all six genes in one go, with efficiencies ranging from ~30% to 80%. Analysis of those mutants identified which EPS genes were critical for capsule formation. The significance of this work is two-fold: (1) It demonstrates multiplex editing using endogenous CRISPR, and (2) It confirms that CRISPR systems present in food-grade LAB (like *S. thermophilus*, used in yogurt) can be repurposed without adding any non-native DNA. This could alleviate regulatory concerns for genome-edited cultures in the food industry, since the edits can, in principle, be made by the bacterium’s own machinery.

More recently, Gu and colleagues reported the first utilization of a native Cas9 (class 2, type II) in a lactobacillus: the endogenous type II-A CRISPR/Cas9 of *Lacticaseibacillus paracasei* was reprogrammed into a high-efficiency editing system [54]. Once its protospacer-adjacent motif (PAM) was characterized and a custom sgRNA cassette introduced, the *L. paracasei* Cas9 achieved gene deletions or insertions at over 90% efficiency, nucleotide substitutions at ≥50% efficiency, and even multi-kilobase deletions (5–10 kb) or simultaneous multi-gene knockouts in one step—feats unprecedented in lactobacilli using any CRISPR platform at the time [54].

Many LAB (e.g., *Lactobacillus fermentum*, *Lactococcus lactis* IL1403) have Class 1 CRISPR systems (like type I or III, which use Cascade or Csm complexes rather than Cas9). These systems are harder to directly repurpose for editing because they do not cause simple DSBs—type I systems cleave DNA in a multi-step process involving a helicase-nuclease (Cas3), and type III target RNA and can cause collateral effects. In 2019, Hidalgo-Cantabrana and colleagues demonstrated the first repurposing of an endogenous class 1 CRISPR/Cas system in lactobacilli by harnessing the native type I-E locus of *Lactobacillus crispatus* [55]. By providing a synthetic CRISPR array and a homologous repair template, they enabled the Cascade–Cas3 machinery to introduce targeted mutations in this previously genetically recalcitrant species—achieving a 643 bp gene deletion with 100% efficiency, as well as a stop-codon insertion (36%) and a single-base substitution (19%) [55]. This pioneering work proved that the abundant but complex multi-protein CRISPR systems in LAB can be exploited for precise genome edits, thereby expanding the genetic toolkit beyond the canonical single-effector nucleases.

Using an endogenous system has advantages: the Cas protein is already optimized for expression and function in that host, and one needs to deliver only a small CRISPR array plus donor DNA. It also means no coding sequence for Cas is integrated or transiently expressed, which can be beneficial for strain safety profiles (particularly in food applications). A limitation is that not all spacers you introduce will be efficiently processed or utilized by the native machinery; sometimes the host may have anti-CRISPRs or other regulators. Additionally, working with endogenous systems requires detailed knowledge of their locus architecture to properly design the CRISPR array plasmids. Together, these foundational studies confirmed the feasibility of leveraging LAB’s own CRISPR/Cas arsenal for genome engineering and laid the groundwork for self-contained CRISPR editing approaches in LAB, by illuminating native PAM requirements, guide processing features, and the potential for scarless, high-efficiency modifications without introducing foreign nucleases.

In conclusion, repurposing LABs’ own CRISPR/Cas systems is an elegant approach that has been successfully demonstrated in a few cases. *P. acidilactici* and *S. thermophilus* now join the ranks of bacteria like *Staphylococcus aureus* and *Francisella novicida*, where endogenous CRISPRs have been harnessed for genome engineering. As CRISPR biology continues to be explored, we may find more endogenous systems in LAB that can be co-opted—including, perhaps, class 1 systems for specialized applications or using endogenous Cas12a if any LAB are found to have type V loci. The ability to leverage what nature provided, rather than importing foreign Cas nucleases, aligns well with the ethos of using LAB as safe organisms (e.g., for food, one might prefer not to introduce *S. pyogenes* genes).

Still, while harnessing endogenous CRISPR/Cas systems (when present in a given LAB strain) can sometimes enable genome editing without introducing foreign Cas enzymes, this strategy is inherently limited—only those few LAB strains that naturally carry active CRISPR/Cas loci can be edited this way, and even then the native systems often have narrow PAM requirements or lower activities compared to *Streptococcus pyogenes* Cas9. In practice, such endogenous tools are confined to specific species and must be painstakingly optimized for each case. Therefore, for the majority of LAB, heterologous CRISPR/Cas systems (like SpCas9 or Cas12a expressed from a plasmid) remain essential to achieve genome edits, given their broader targeting range and proven reliability.

These efforts also deepen our fundamental understanding of LAB CRISPR systems, as engineering them often reveals their targeting rules, efficiencies, and any quirks (for instance, spacer size preferences, requirement for a specific tracrRNA, etc.). The synergy of endogenous and exogenous CRISPR tools gives LAB researchers a rich toolbox to choose from.

## 4. Conclusions

LAB genome-editing has progressed from niche, low-throughput techniques to a diversified toolkit rivaling that of traditional model organisms. Native RecA pathways remain valuable for large (>10 kb) integrations where high fidelity outweighs speed, while phage-mediated recombineering delivers rapid, high-efficiency gene replacements. CRISPR/Cas counter-selection transforms sub-percent recombination events into near-quantitative outcomes and when coupled with oligonucleotide donors, enables scar-less, marker-free point mutations. Endogenous Cas systems demonstrate that sophisticated edits can be achieved without foreign nuclease genes, which is an advantage for food-grade and probiotic applications. The emergence of CRISPR-associated transposases overcomes size constraints, foreshadowing streamlined pathway transplantation.

Key challenges ahead include expanding recombineering host-range, minimizing off-target Cas activity, integrating multiplex edits without cumulative scars, and devising universal, non-antibiotic selection schemes. Addressing these gaps will accelerate the deployment of LAB as modular cell factories and live biotherapeutic vehicles, fulfilling the promise envisioned as the field moves “from past, through present, to future”.

## Figures and Tables

**Figure 1 ijms-26-07483-f001:**
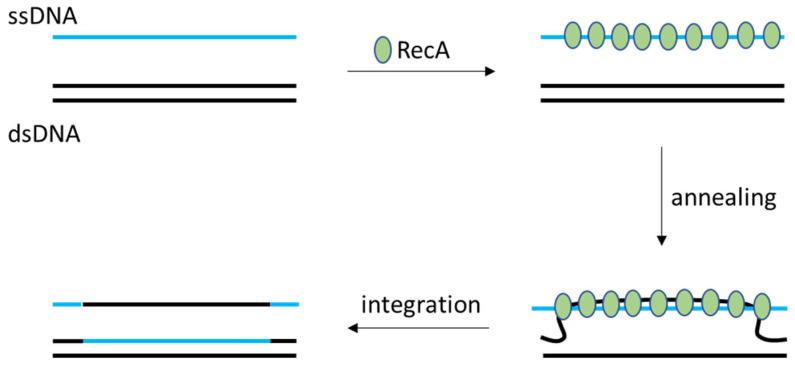
Schematic RecA-mediated recombination. RecA proteins help anneal ssDNA to the chromosome or plasmid, forming triple-stranded DNA complexes, after which ssDNA fragments integrate into dsDNA.

**Figure 2 ijms-26-07483-f002:**
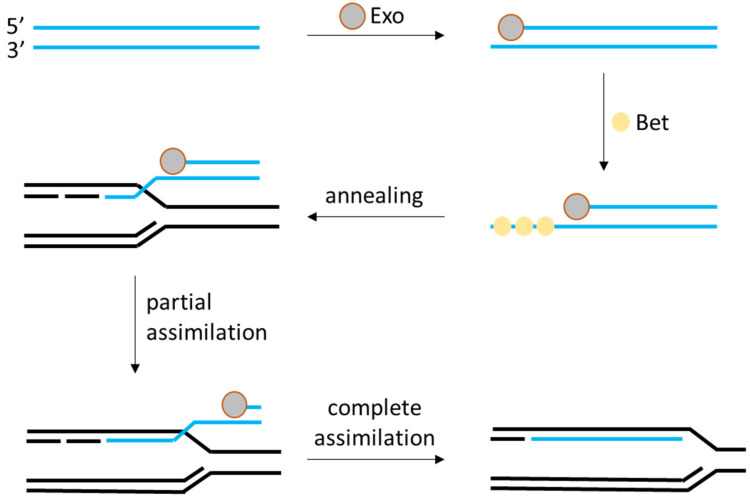
λ-Red-mediated integration of foreign DNA. Exo protein resects dsDNA, Bet protein binds and anneals the resulting ssDNA to the lagging strand of the replicative fork, and Gam protein inhibits host’s nucleases protecting the ssDNA. After annealing foreign ssDNA assimilates with the host’s genome or plasmid DNA resulting in gene disruption or integration depending on the DNA inserted.

**Figure 3 ijms-26-07483-f003:**
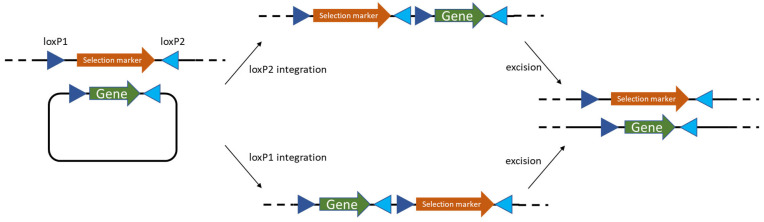
Target gene integration into genomic DNA via Cre/loxP system. For more precision two different loxP sites are needed. During the final loxP excision target gene will be integrated into the genome half of the time and cells that had this swap occur can be further enriched for or screened in the absence of the selective marker.

**Figure 4 ijms-26-07483-f004:**
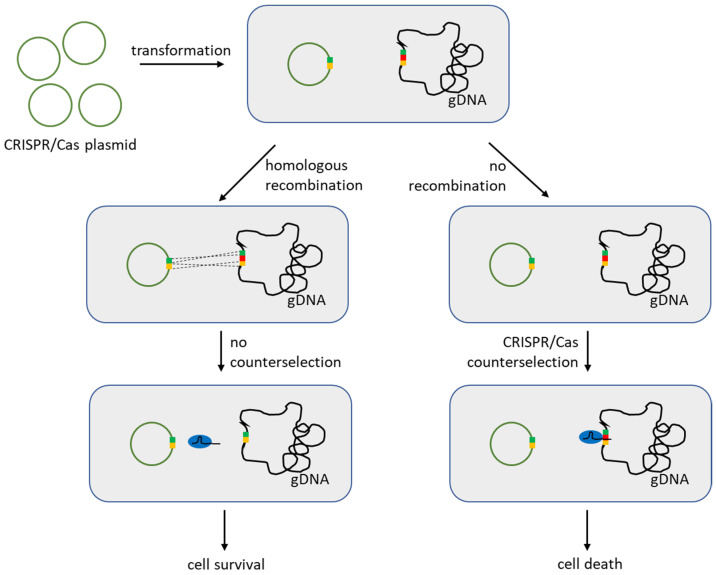
Principle of CRISPR/Cas counter-selection. Plasmid with CRISPR/Cas contains homologous to the gDNA regions which flank target gene. If homologous recombination occurs, then target gene region on the gDNA mutates (either by removing target gene entirely or by mutating it otherwise) in which case Cas protein cannot cut gDNA and the cell survives. If recombination does not occur, then Cas protein simply cuts gDNA in the target region and the cell dies. Cas9 is shown as blue circle with black sgRNA inside. Red rectangle shows the recognition site of Cas9 on the gDNA. Green and yellow rectangles show the adjacent parts of gDNA that can recombine with the plasmid.

**Figure 5 ijms-26-07483-f005:**
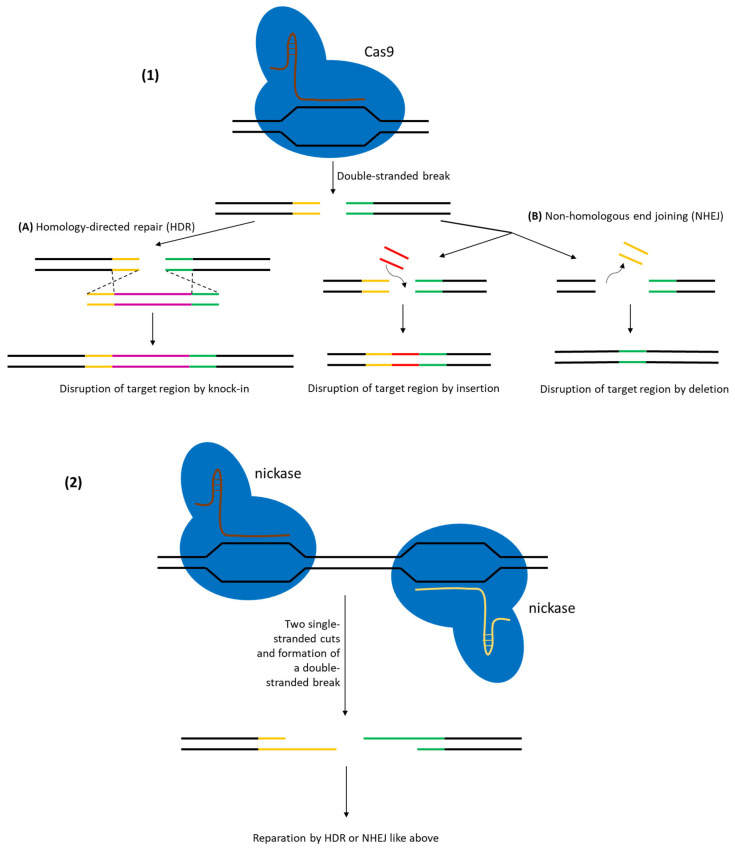
DNA repair after (**1**) CRISPR/Cas cut or (**2**) double nickase cut. (**A**) shows a homology-directed repair which can create precise knock-ins or knockouts depending on the donor DNA. (**B**) shows a non-homologous end joining by either a deletion or an insertion which can disrupt the target region and even the whole locus.

**Figure 6 ijms-26-07483-f006:**
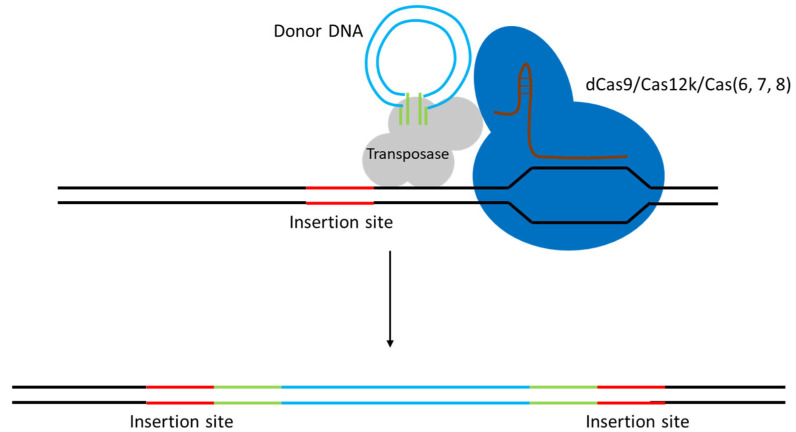
Efficient large DNA fragment insertion via a CRISPR-associated transposon. Using catalytically active Cas9 can lead to errors due to DSB creation at the recognition site, which is why it is beneficial to use either dCas9 or different Cas proteins (such as Cas12k or Cas6) to increase the success rate of transposition.

**Figure 7 ijms-26-07483-f007:**
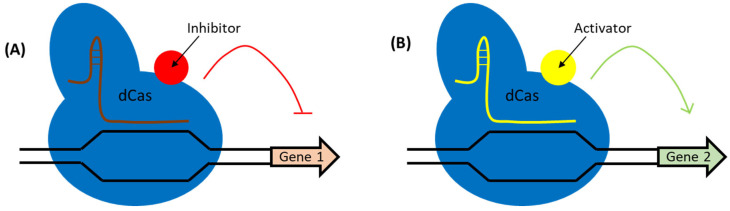
CRISPRi and CRIPSRa systems. (**A**) catalytically dead Cas protein is fused with the inhibitor (or transcription repressor; although in some cases repression is achieved via steric hindrances without inhibitor domain) and when is bound to the target region it cannot cleave it inhibits the transcription of the target gene. (**B**) catalytically dead Cas protein is fused with the transcription activator and when is bound to the target region it cannot cleave it activates the transcription of the target gene.

**Table 1 ijms-26-07483-t001:** Comparison of genome-editing tools for LAB. Editing efficiencies and insert sizes are approximate, based on the current literature.

Genome-Editing Tool/Approach	Nuclease(s) Employed	Editing Efficiencies Achieved	Validated Host Strains	Key Citation(s)
**RecA-mediated double-crossover** (classical homologous recombination)	None (RecA-dependent HR only)	Very low (~10^−6^–10^−5^ per cell without selection). Marker selection or counter-selection is usually required to recover mutants.	*L. lactis*, *L. plantarum*, *L. casei* (broadly all LAB, but extremely low efficiency without selection)	[2]
**Phage RecT/RecE recombineering** (λ-Red analogs from LAB prophages)	None (phage-encoded recombinases, e.g., RecT/Beta and RecE/Exo)	High for single-gene edits: often >50% of colonies correct; up to ~100% in optimized cases (e.g., 167 bp deletion or GFP insertion). Larger edits (~5 kb insertions) feasible with selection	*L. plantarum* WCFS1, *L. casei* BL23, *L. lactis* (with heterologous RecT). Primarily species-specific systems (phage origin) but functional across some LAB genus	[15,16]
**CRISPR–Cas9 (DSB with HDR template)** (Cas9-mediated counter-selection)	SpCas9 (wild-type, DSB-forming)	Point mutations and small deletions: ~90–100% efficiency (near-complete editing in *L. reuteri* with oligo donors). Small insertions (≤100 bp): ~50–75% efficiency in one step. Larger inserts (~1–2 kb) achieved at high frequency with optimized two-plasmid systems (e.g., ~80–90% for ~1 kb insert)	*L. reuteri* ATCC 6475, *L. lactis* NZ9000, *L. plantarum* WJL, *L. paracasei*, *L. plantarum* WCFS1, others (many LAB with species-specific optimization)	[34,35]
**CRISPR–nCas9 (nickase-mediated HDR)** (Cas9-D10A single-strand nick)	SpCas9 D10A (nickase variant)	25–62% efficiency for precise deletions/insertions in *L. casei*. Up to ~60% efficiency in various *Lactobacillus* species using a portable nCas9 system. Significantly improved cell survival compared to DSB-causing Cas9	*L. casei* BL23, *L. acidophilus*, *L. gasseri*, *L. paracasei* (demonstrated across multiple probiotic *Lactobacillus*)	[37,38]
**CRISPR base editors** (dCas9/nCas9 fused to deaminase for C→T or A→G editing)	Cas9 nickase or dCas9 fused with cytidine deaminase (CBE) or adenine deaminase (ABE)	High efficiency single-nucleotide conversions without DSB or donor DNA. For example, ~80–100% of cells acquired target C→T or A→G mutations in *L. lactis*, and similarly high efficiencies were achieved at multiple loci simultaneously. In *L. plantarum*, a multiplexable base-editing (Target-AID) system showed efficient point mutations with no survival penalty	*L. lactis* NZ9000, *L. plantarum* WCFS1, other *Lactobacillus* spp. (feasibility shown in multiple strains)	[39,41]
**CRISPR-guided transposase** (CAST; CRISPR-associated transposon integration)	Type I-F Cascade (Cas6/7/8 complex, no cutting) + TnsABC transposase (Tn7-like)	Allows large DNA insertions without HDR. ~2 × 10^−4^ efficiency for ~1 kb insertions, and on the order of 10^−5^ for ~10 kb payloads in *L. lactis*. Notably, inserts up to 10 kb were stably integrated in one step. Currently low absolute efficiency (10^−4^–10^−5^), but a major advance for payload size	*L. lactis* MG1363 (first demonstrated in LAB). Species-specific (system requires retooling for each host)	[45]

## Data Availability

Only publicly available datasets were analyzed in this study.

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
