# Peer review of "Genome-Editing Tools for Lactic Acid Bacteria: Past Achievements, Current Platforms, and Future Directions"

_ijms, 2025, doi:10.3390/ijms26157483_

Round 1

Reviewer 1 Report

Comments and Suggestions for Authors

Dear Authors!

Lactic acid bacteria (LAB) are a very important for the food and pharmaceutical industries, but their genomes are very difficult to edit. The authors of the Review spent considerable time analyzing and summarizing the relevant references, as can be seen from the structure of the review and the description of LAB genome editing methods.

However, in the presented Review there is no mention of any articles on the subject that were published by the authors. It is unclear whether the authors conducted relevant research on this topic. If so, it would be helpful to mention this information in the introduction or other parts of the Review to increase the credibility and citation of the work.

The authors analyzed 54 scientific articles, mostly recent publications. They did not specify the databases they used or the criteria they applied in selecting the documents. To improve the methodology of this research, it is suggested to include a description of the search strategy used.

Additionally, there are some minor comments on the use of abbreviations in the text. For instance, it was enough to mention once at the beginning of the text that LAB means ‘Lactic acid bacteria’. This interpretation is used several times throughout the Review (pages 1, 2, 9).

After considering these comments, this Review can be recommended for publication in International Journal of Molecular Sciences with minor revisions.

Author Response

Dear Reviewer!

We sincerely thank you for positive feedback of our manuscript.

Comment 1: However, in the presented Review there is no mention of any articles on the subject that were published by the authors. It is unclear whether the authors conducted relevant research on this topic. If so, it would be helpful to mention this information in the introduction or other parts of the Review to increase the credibility and citation of the work.

Response 1: We thank reviewer for this comment. While we indeed are working in the field of LAB microbiology and engineering, this particular Review was intended to provide an objective and comprehensive summary of published methods by various research groups. Currently our team is improving our competences on LAB genetic engineering. Experimental research articles on this theme will be published later so we cannot cite them right now in this Review.

Comment 2: The authors analyzed 54 scientific articles, mostly recent publications. They did not specify the databases they used or the criteria they applied in selecting the documents. To improve the methodology of this research, it is suggested to include a description of the search strategy used.

Response 2: We thank reviewer for this comment. We’ve made appropriate changes in the Introduction section describing the search strategy for this review article (highlighted in cyan).

Comment 3: Additionally, there are some minor comments on the use of abbreviations in the text. For instance, it was enough to mention once at the beginning of the text that LAB means ‘Lactic acid bacteria’. This interpretation is used several times throughout the Review (pages 1, 2, 9).

Response 3: We thank reviewer for this comment. We’ve decided to leave two abbreviation translations of LAB in the Abstract and at the very beginning of Introduction. We’ve removed other repetitions throughout the text (highlighted in green).

Reviewer 2 Report

Comments and Suggestions for Authors

This manuscript provides a comprehensive and timely overview of the evolving genome-editing toolkit for lactic acid bacteria (LAB), highlighting key advancements from classical homologous recombination to cutting-edge CRISPR-based systems. The authors effectively synthesize a broad range of techniques, emphasizing how innovations like RecT/RecE recombineering, CRISPR-Cas counter-selection, and CRISPR-associated transposases have addressed historical limitations (e.g., low efficiency, marker scars). This review bridges fundamental techniques and emerging trends, making it a valuable resource for microbiologists and metabolic engineers.

Minor suggestion:

  1. For Part 3.2:

It is recommended to relocate the content on 'Ma et al. (2024) managed to target...' to Part 3.5, as it pertains to endogenous CRISPR/Cas systems. Additionally, the section discussing 'Though not a permanent mutation, CRISPR interference...' should be moved to Part 3.4, which focuses on CRISPR interference (CRISPRi). This reorganization would enhance the logical flow and clarity of the review.

  1. For Part 3.5:

The section should include a supplementary discussion on the earliest characterized type I and type II endogenous CRISPR-Cas systems in lactobacilli (see PMID: 31341082 and 36414383), as these foundational studies are critical for understanding the development of CRISPR-based tools in lactic acid bacteria (LAB).

Author Response

Dear Reviewer!

We sincerely thank you for positive feedback of our manuscript.

Comment 1: For Part 3.2:

It is recommended to relocate the content on 'Ma et al. (2024) managed to target...' to Part 3.5, as it pertains to endogenous CRISPR/Cas systems. Additionally, the section discussing 'Though not a permanent mutation, CRISPR interference...' should be moved to Part 3.4, which focuses on CRISPR interference (CRISPRi). This reorganization would enhance the logical flow and clarity of the review.

Response 1: We thank reviewer for this comment. We’ve made appropriate changes to the text (highlighted in green). The text from the first part of this comment was deleted entirely since the Part 3.5 already contains well-integrated discussion of the aforementioned study by Ma et al.

Comment 2: For Part 3.5:

The section should include a supplementary discussion on the earliest characterized type I and type II endogenous CRISPR-Cas systems in lactobacilli (see PMID: 31341082 and 36414383), as these foundational studies are critical for understanding the development of CRISPR-based tools in lactic acid bacteria (LAB).

Response 2: We thank reviewer for this comment and for highlighting additional articles helpful to this Review paper. We’ve made appropriate changes to the text (highlighted in cyan).